# The interaction of ammonia and manganese in abnormal metabolism of minimal hepatic encephalopathy: A comparison metabolomics study

Xue-Fei Liu[☯], Jing-Jing Lu[☯], Ying Li[ID]*, Xiu-Ying Yang, Jin-Wei Qiang*

Department of Radiology, Jinshan Hospital, Fudan University, Shanghai, China

☯ These authors contributed equally to this work.
* dr.yingli@foxmail.com (YL); dr.jinweiqiang@163.com (JWQ)

**Data Availability Statement:** The data underlying the results presented in the study are available from https://gitee.com/dr_yingli/mn-mrs.

## Abstract

This study was to investigate the effects of ammonia and manganese in the metabolism of minimal hepatic encephalopathy (MHE). A total of 32 Sprague-Dawley rats were divided into four subgroups: chronic hyperammonemia (CHA), chronic hypermanganese (CHM), MHE and control group (CON). [1]H-NMR-based metabolomics was used to detect the metabolic changes. Sparse projection to latent structures discriminant analysis was used for identifying and comparing the key metabolites. Significant elevated blood ammonia were shown in the CHA, CHM, and MHE rats. Significant elevated brain manganese (Mn) were shown in the CHM, and MHE rats, but not in the CHA rats. The concentrations of γ-amino butyric acid (GABA), lactate, alanine, glutamate, glutamine, threonine, and phosphocholine were significantly increased, and that of myo-inositol, taurine, leucine, isoleucine, arginine, and citrulline were significantly decreased in the MHE rats. Of all these 13 key metabolites, 10 of them were affected by ammonia (including lactate, alanine, glutamate, glutamine, myo-inositol, taurine, leucine, isoleucine, arginine, and citrulline) and 5 of them were affected by manganese (including GABA, lactate, myo-inositol, taurine, and leucine). Enrichment analysis indicated that abnormal metabolism of glutamine and TCA circle in MHE might be affected by the ammonia, and abnormal metabolism of GABA might be affected by the Mn, and abnormal metabolism of glycolysis and branched chain amino acids metabolism might be affected by both ammonia and Mn. Both ammonia and Mn play roles in the abnormal metabolism of MHE. Chronic hypermanganese could lead to elevated blood ammonia. However, chronic hyperammonemia could not lead to brain Mn deposition.

## Introduction

Hepatic encephalopathy (HE) refers to a central nervous system dysfunction caused by liver dysfunction and/or portosystemic shunting [1]. Minimal hepatic encephalopathy (MHE) is the early stage of HE, which has no typical clinical manifestations, but has abnormal in

**Funding:** The results reported herein correspond to specific aims of grant No. 2021-3-01 to investigator Ying LI from Jinshan Science and Technology Committee. This work was also supported by grant No. ZK2019B01 to investigator Jin-Wei QIANG from Shanghai Municipal Health Commission. The funders had no role in study design, data collection and analysis, decision to publish, or preparation of the manuscript.

**Competing interests:** The authors have declared that no competing interests exist.

cognitive behavior [2]. The pathogenesis of MHE is complex and has not been fully elucidated. Presently, three major theories including ammonia poisoning, branched chain amino acids (BACCs) imbalance, and γ-amino butyric acid (GABA) neurotransmitter abnormality are considered as consensus [1, 2].

Due to liver dysfunction, elevated blood ammonia is commonly seen in type A (caused by acute liver dysfunction) and type C (The most common MHE, caused by both liver dysfunction and portosystemic shunting) MHE patients. However, previous studies showed that the liver function and blood ammonia remains normal in type B MHE (caused by portosystemic shunting) patients. Furthermore, elevated blood manganese (Mn) and brain Mn deposition were reported, due to the presence of portosystemic shunting and bypassing of liver excretion of Mn [3]. Mn is neurotoxic, which can lead to amino acid metabolic imbalance and impaired GABAergic activity [4, 5]. Reducing Mn intake can significantly improve the cognitive state of MHE rats [6]. These evidences indicated that Mn may play an important role in the abnormal metabolism of MHE.

Nuclear magnetic resonance spectroscopy (NMR) is a technique for continuous observation metabolites include amino acids and fatty acids, which can be found in the blood, urine, and other bodily fluids, and tissues [7]. Recently, with the development of metabolomics analysis technology, metabolomics has attracted increasing attention in the medical field as a method to search for metabolism biomarkers for diseases [8].

We assumed that both ammonia and Mn play a role in the abnormal metabolism of MHE with interactions between these two toxic substances. To investigate the effects of ammonia and Mn in the metabolism of MHE, $^1$H-NMR-based metabolomics was used to detect the metabolic changes in an MHE rat model. And the metabolic changes were compared with that of a chronic hyperammonemia rat model and a chronic hypermanganes rat model. We expected to clarify the effect of ammonia and Mn in the abnormal metabolism of MHE.

## Materials and methods

### Research ethics

Ethical approval for this study was obtained from the Institutional Review Board of Jinshan Hospital, Fudan University (No. 2017–01). The present study followed Institutional Animal Care and Use Committee of Fudan University guidelines for humane animal treatment and complied with relevant legislation.

### Laboratory animals and models

Six weeks male Sprague-Dawley rats (weighing about 150 g) were divided into four subgroups: chronic hyperammonemia (CHA, n = 8), chronic hypermanganese (CHM, n = 8), MHE (n = 8) and controls (CON, n = 8). The CHA rats were given oral administration of 30 mg/kg/day ammonium acetate (150 mg of ammonium acetate dissolved in 500 mL of deionized water) [9]. The CHM rats were given oral administration of 30 mg/kg/day manganese chloride (150 mg of manganese chloride dissolved in 500 mL of deionized water) [10]. The MHE rats were given oral administration of 100 mg/kg/day (TAA) thioacetamide (500 mg TAA dissolved in 500 mL of deionized water) [11]. All the doses were adjusted weekly according to the rats' body weight and water consumption. The CON rats were given deionized water without any treatment. The rats were placed in metabolic cages (one/cage) and housed in a well-ventilated room with temperature (25±2˚C), air humidity (50±10%) and a light/dark cycle of 12 h. All the rats were allowed to drink and eat freely for 8 weeks. The feeds were prepared according to the ain-93 experimental animal rat purified feed standard.

## Neurological function assessment

Morris water maze (MWM) were used for cognitive assessment at the end of 8 weeks. A black round pool with 180 cm diameter and 50 cm height were filled with water (30 cm from the bottom). The water temperature is adjusted to 24±1˚C. The pool is divided into four quadrants (left top, right top, left bottom, and right bottom). A circular transparent platform of 12 cm was placed in the right top with 2 cm below the water surface. The rats were randomly placed from one quadrant into the water. Each rat had 60 s to find the platform. the time of the rats climbed the platform was record as the latency. If the rats were unable to reach the platform within 60 s, they would be guided to the platform to rest for 20 s. All the rats were trained for 4 consecutive days. A test trial of MWM was performed with the platform withdrawn, and the number of times a rat's head reach the platform's original location was recorded. The tracks of the rats were recorded by a computer equipped with a camera.

Narrow beam walking test was used to evaluate the locomotion of the rats. The time to cross the start line (latency time) and the time to cross the beam (total time) were recorded.

## Sample preparation, blood ammonia and brain Mn content measurements

After the neurological function assessment, the rats were sacrificed by cervical dislocation under deep anesthesia of isoflurane. 1 mL of blood was drawn from the ventriculus dexter for assessing blood ammonia. The brain tissue was removed, cleaned by cold saline, and died by filter papers. The striatum was separated on a frozen platea ccording to the anatomical map of rat brain. The samples were stored in liquid nitrogen and then transferred to a -80˚C refrigerator. The brain Mn content was determined by using inductively coupled plasma spectrometer.

## Enzyme activity measurements

Commercial kits were used to analyze the activity of glutamine synthetase (GS) and pyruvate carboxylase (PC). The samples prepared and proceeded according to the protocols (http://www.njjcbio.com). The results were expressed as units of enzyme activity per mg protein (U/mg).

## $^1$H-NMR metabolomics

Samples were weighed and added with ice-cooled extraction solution ($CH_3OH$:$H_2O$ = 2:1). The sample concentration was 100 mg/mL. After vortexing for 1 min, the sediments were broken by a tissue crusher (30 Hz, 90 s). After ice bath for 2 min, the supernatant was centrifuged (4˚C, 10,000×g, 10 min), rotary evaporated, and freeze-dried for 24 h. The solid residues were added with 600 μL buffer solution (0.1 mL NaH2PO4/K2HPO4, pH: 7.4 with 99.96% deuteriumenriched $D_2O$ and 0.05% TSP). The TSP was used as a chemical shift reference. Then samples were then transferred into 5 mm $^1$H-NMR tubes.

The $^1$H-NMR was performed by a 600 MHz 9.4 T NMR spectrometer (AVANCE III, Bruker, German) at 298 K with Carr-Purcell-Meiboom-Gill (CPMG) pulse sequence and water suppression. The basic parameters were as follows: water presaturation during the relaxation delay = 2.5 s; pulse = 90˚; width = 10 ms; sampling points = 32 K; sampling time = 1.36 s; scanning times = 64; spectral width = 20 ppm.

$^1$H-NMR free induction decays data were processed in a previous reported pipeline [12]. Briefly, group delay correction, solvent suppression, apodization, fourier transform, zero-order phase correction, internal referencing, baseline correction, negative values zeroing, warping, window selection, bucketing, water region removal, zone aggregation and normalization were performed in sequences. The identification and quantification of the metabolites

were performed by using a spectroscopy of 191 pure metabolites as a reference. Metabolites within 2 standard deviations of the background noise were not included in the further analysis.

Supervised analysis (sparse projection to latent structures discriminant analysis, sPLS-DA with 3-fold cross-validation) was used for pattern recognition analysis of the significantly affected metabolites in CHA vs. CON, CHM vs. CON, and MHE vs. CON rats. The metabolites with variable influence on projection (VIP) score > 1 and P < 0.05 were defined as the key metabolites. The identical metabolites found both in CHA vs. CON and MHE vs. CON were thought as the metabolomics signatures in MHE affected by the ammonia. The identical metabolites found both in CHM vs. CON and MHE vs. CON were thought as the metabolomics signatures in MHE affected by the Mn.

A diffusion algorithm was used for enrichment analysis of the key metabolomics in MHE and in the metabolomics signatures of ammonia and Mn. The top-ranked significant different biological pathways of CHA vs. CON, CHM vs. CON, and MHE vs. CON rats were also represented based on the KEGG database (https://www.kegg.jp/).

## Statistical analysis

All statistical analyses were performed in R (Version 4.0.2; http://www.r-project.org/); the "mixOmics" package was used for sPLS-DA analysis and VIP score calculation; the "FELLA" package was used for enrichment and pathway analysis. Data met normality and variance homogeneity were analyzed by ANOVA followed by false discovery rate (FDR) correction. If not met normality or variance homogeneity, the data were analyzed by Mann-Whitney U test followed by FDR. P < 0.05 was considered statistically significant.

## Results and discussion

### General situation

Significant elevated blood ammonia, and prolonged escape latency of MWM were seen in the CHA, CHM, and MHE rats. Brain Mn elevation were shown in CHM, and MHE rats, but not in the CHA rats (Table 1).

Table 1. The laboratory and neurological characteristics of the rat models.

|  | CON (n = 8) | CHA (n = 8) | CHM (n = 8) | MHE (n = 8) |
|---|---|---|---|---|
| Blood ammonia (μmol/L) | 131 (24.0) | 161 (41.3)* | 150 (30.3) | 206 (9.9)*** |
| Mn (ug/g wet tissue) | 0.84 (0.24) | 0.74 (0.20) | 1.52 (0.48)** | 1.15 (0.32)* |
| Glutamine synthetase (U/mg) | 1.4 (0.3) | 2.6 (0.4)*** | 1.6 (0.3) | 2.2 (0.1)*** |
| Pyruvate carboxylase (U/mg) | 2.3 (0.3) | 2.9 (0.6)* | 2.9 (0.7)* | 3.6 (0.1)*** |
| Test 1 (s) | 60.0 (0.4) | 59.7 (0.7) | 58.9 (1.2) | 59.8 (0.4) |
| Test 2 (s) | 49.7 (9.8) | 57.7 (2.3) | 55.1 (2.2) | 52.2 (6.6) |
| Test 3 (s) | 27.4 (17.0) | 49.9 (5.5)** | 52.3 (8.5)** | 43.7 (13.4)* |
| Test 4 (s) | 19.2 (4.41) | 42.1 (9.2)*** | 43.6 (5.3)*** | 42.0 (7.7)*** |
| Times | 2.8 (0.8) | 1.8 (0.6)* | 2.0 (0.7)* | 0.7 (0.4)*** |
| Latency (s) | 6.8 (1.9) | 17.3 (2.4)*** | 17.8 (4.2)*** | 22.5 (2.8)*** |
| Total time (s) | 16.4 (3.2) | 25.3 (4.2)*** | 26.8 (6.6)*** | 27.8 (8.3)** |

*, P < 0.05

**, P < 0.01

***, P<0.001, compared with CON.

## Metabolites identification and quantification

The $^1$H-NMR spectroscopy of CHA, CHM, MHE, and CON rats showed that 45 metabolites were identified for further key metabolites analysis. All the 45 metabolites were shown in a heatmap with hierarchical clustering (Fig 1).

## Metabolomics signatures selection and validation

The metabolites between CHA and CON rats ($R^2x = 0.20$, $R^2y = 0.96$, $Q^2 = 0.80$), between CHM and CON rats ($R^2x = 0.40$, $R^2y = 0.94$, $Q^2 = 0.69$), and between MHE and CON rats ($R^2x = 0.66$, $R^2y = 0.99$, $Q^2 = 0.85$) could be well separated by sPLS-DA (Fig 2).

Fourteen significantly affected metabolites were identified between CHA and CON rats. The concentrations of lactate, alanine, glutamate, glutamine, and glycerol were significantly increased, while that of myo-inositol, taurine, acetic acid, leucine, isoleucine, valine, arginine, citrulline and citrate were significantly decreased in CHA rats. Fourteen significantly affected metabolites were identified between CHM and CON rats. The concentrations of GABA, lactate, and glycerol were significantly increased, while that of taurine, leucine, creatine, myo-inositol, glycine, serine, acetic acid, valine, malic acid, cysteine, and threonic acid were significantly decreased in CHM rats. Thirteen significantly affected metabolites were identified between MHE and CON rats. The concentrations of GABA, lactate, alanine, glutamate, glutamine, threonine, and phosphocholine were significantly increased, while that of myo-inositol, taurine, leucine, isoleucine, arginine, and citrulline were significantly decreased in MHE rats.

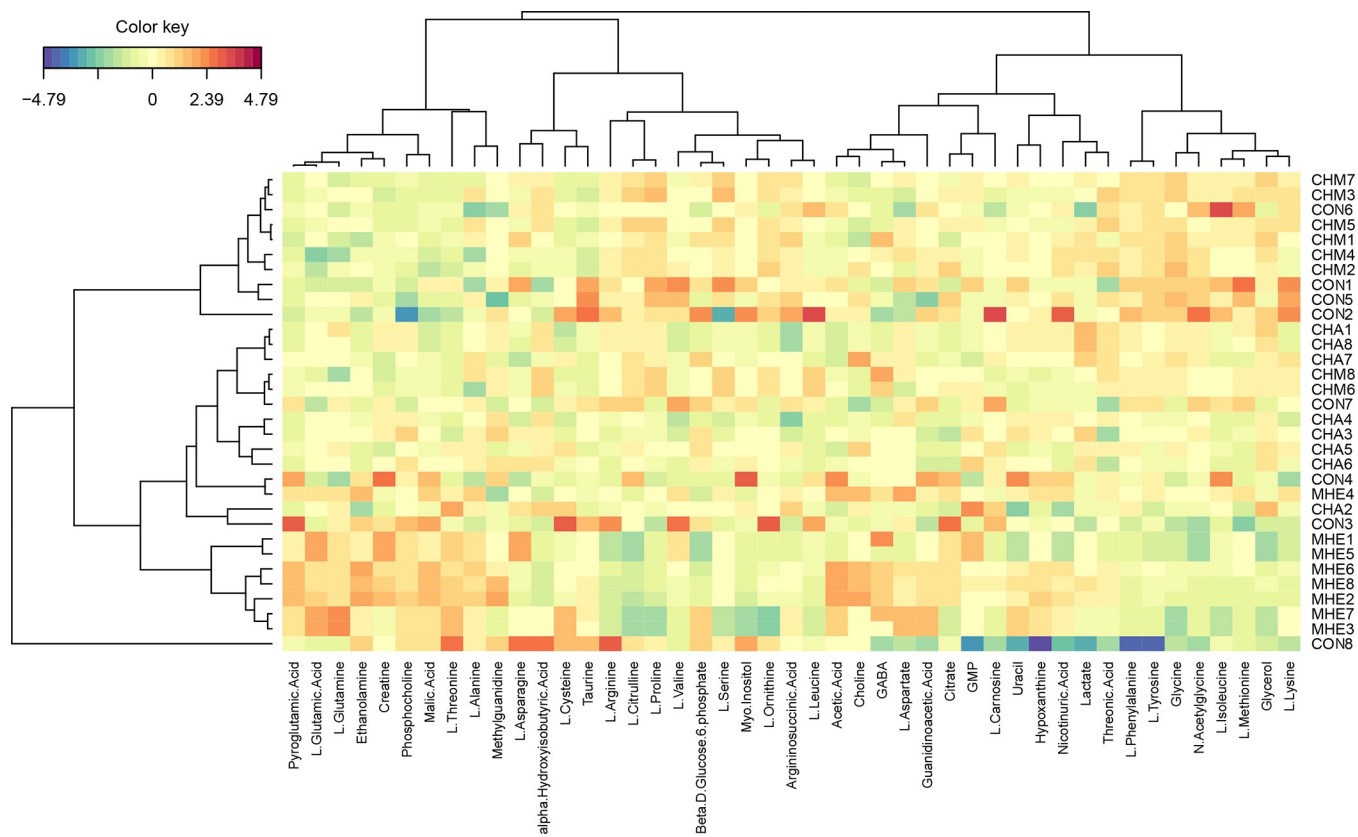

**Fig 1. Two-dimensional heatmap from hierarchical clustering shows all the 45 metabolites in the striatum in the CHA, CHM, MHE, and CON rats.**

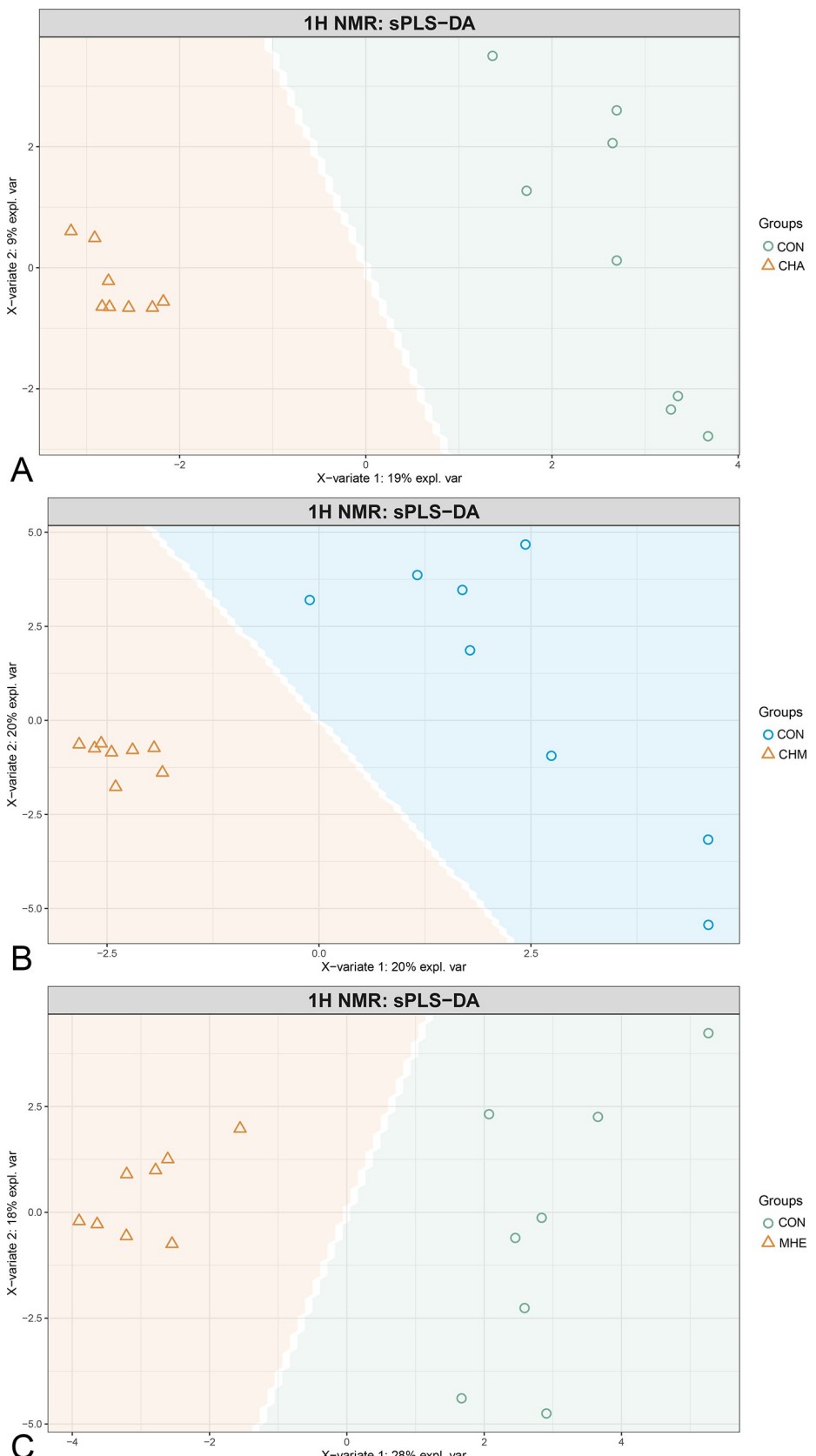

**Fig 2. Sparse projection to latent structures discriminant analysis (sPLS-DA) of the metabolites.** The metabolites between CON and CHA rats (A), CON and CHM rats (B), and CON and MHE rats (C) could be separated by sPLS-DA.

Ten metabolomics signatures were identified involved both in CHA and MHE rats, namely, lactate, alanine, glutamate, glutamine, myo-inositol, taurine, leucine, isoleucine, arginine, and citrulline. Five metabolomics signatures were identified involved both in CHM and MHE rats, namely, GABA, lactate, myo-inositol, taurine, and leucine (Table 2) (Fig 3).

## Enrichment and pathway analysis

The pathway analysis of the thirteen key metabolites from MHE shows that of the arginine metabolism, alanine, aspartate and glutamate metabolism, glycine, serine and threonine metabolism, valine, leucine and isoleucine metabolism, arginine and proline metabolism, taurine and hypotaurine metabolism, pyruvate metabolism, aminoacyl-tRNA metabolism, and GABAergic synapse is evolved in the MHE metabolism (S1 Table). That of the arginine metabolism, alanine, aspartate and glutamate metabolism, valine, leucine and isoleucine metabolism, arginine and proline metabolism, taurine and hypotaurine metabolism, and pyruvate metabolism is affected by ammonia in MHE (S2 Table). That of the galactose metabolism, beta-

**Table 2. The fold change of the main metabolites in CHA, CHM, and MHE rats with P values and VIP score.**

| Metabolites | CHA (n = 8) | CHM (n = 8) | MHE (n = 8) | P# | P$ | P* | VIP# | VIP$ | VIP* |
|---|---|---|---|---|---|---|---|---|---|
| Myo-inositol | 0.85 | 0.86 | 0.73 | 0.008 | 0.003 | 0.002 | 1.3 | 1.5 | 1.9 |
| Lactate | 1.42 | 1.31 | 1.25 | 0.018 | 0.005 | 0.003 | 2.1 | 2.1 | 1.1 |
| Glutamine | 1.35 | 0.94 | 1.84 | 0.003 | 0.002 | 0.002 | 1.4 | 0.0 | 2.0 |
| Alanine | 1.21 | 1.11 | 1.31 | 0.003 | 0.002 | 0.002 | 1.3 | 0.2 | 1.6 |
| Acetic acid | 0.87 | 0.92 | 1.07 | 0.003 | 0.002 | 0.002 | 1.9 | 1.5 | 0.0 |
| Taurine | 0.75 | 0.66 | 0.82 | 0.004 | 0.002 | 0.002 | 1.9 | 2.6 | 1.1 |
| Leucine | 0.77 | 0.88 | 0.77 | 0.003 | 0.002 | 0.002 | 2.0 | 1.0 | 1.7 |
| Glutamic acid | 1.24 | 0.99 | 1.72 | 0.003 | 0.002 | 0.002 | 1.6 | 0.0 | 2.4 |
| Glycerol | 1.11 | 1.10 | 0.94 | 0.003 | 0.002 | 0.002 | 1.9 | 2.2 | 0.7 |
| Arginine | 0.86 | 0.92 | 0.78 | 0.003 | 0.002 | 0.002 | 1.5 | 1.0 | 1.9 |
| Valine | 0.86 | 0.91 | 0.92 | 0.002 | 0.002 | 0.002 | 1.7 | 1.3 | 0.5 |
| Isoleucine | 0.87 | 0.93 | 0.86 | 0.002 | 0.002 | 0.002 | 1.6 | 0.9 | 1.3 |
| Citrate | 0.87 | 1.07 | 0.42 | 0.002 | 0.002 | 0.002 | 1.1 | 0.6 | 2.6 |
| Butyrate | 0.71 | 0.93 | 0.97 | 0.002 | 0.002 | 0.002 | 1.5 | 0.0 | 0.0 |
| GABA | 1.05 | 1.11 | 1.15 | 0.017 | 0.005 | 0.004 | 0.8 | 1.9 | 1.8 |
| Creatine | 0.98 | 0.96 | 1.03 | 0.008 | 0.003 | 0.003 | 0.0 | 1.3 | 0.0 |
| Glycine | 1.01 | 1.07 | 0.95 | 0.004 | 0.003 | 0.002 | 0.0 | 1.4 | 0.3 |
| Serine | 0.99 | 1.06 | 0.98 | 0.004 | 0.002 | 0.002 | 0.0 | 1.2 | 0.0 |
| Malic acid | 0.87 | 0.79 | 1.21 | 0.002 | 0.002 | 0.002 | 0.4 | 1.2 | 0.6 |
| Cysteine | 0.83 | 0.78 | 0.91 | 0.002 | 0.002 | 0.002 | 0.9 | 2.0 | 0.0 |
| Threonic acid | 1.20 | 1.69 | 1.28 | 0.002 | 0.002 | 0.002 | 0.0 | 1.3 | 0.0 |
| Threonine | 1.01 | 0.87 | 1.26 | 0.003 | 0.002 | 0.002 | 0.0 | 0.7 | 1.1 |
| Phosphocholine | 1.12 | 1.03 | 1.15 | 0.002 | 0.002 | 0.002 | 1.0 | 0.0 | 1.0 |

#, CHA vs. CON

$, CHM vs. CON

*, MHE vs. CON.

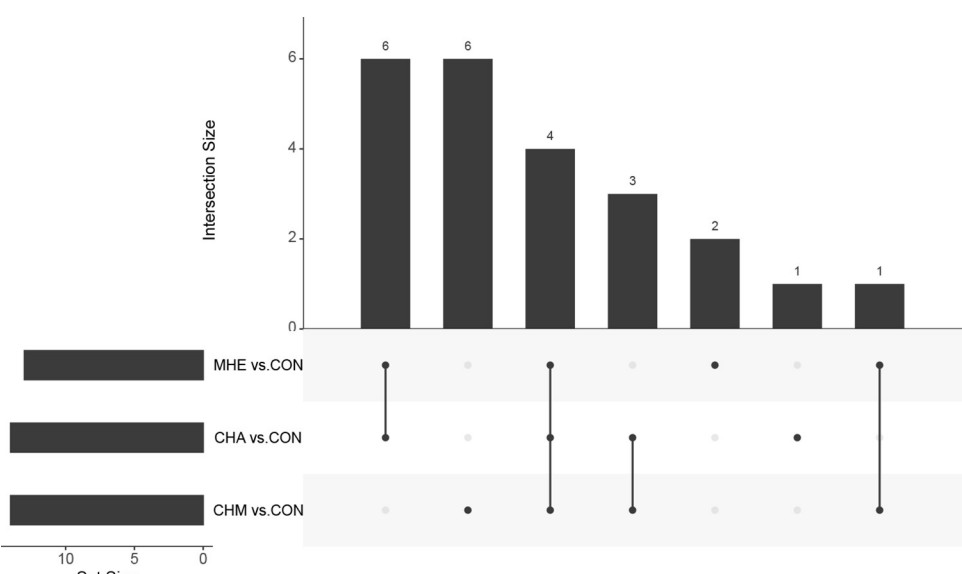

**Fig 3. The metabolomics signatures involved in the CHA, CHM, and CHM rats compared with the controls.** Six metabolomics signatures were identified involved both in CHA and MHE rats, but not in CHM rats. One metabolomics signature was identified involved both in CHM and MHE rats, but not in CHA rats. Four metabolomics signatures were identified involved in CHA, CHM and MHE rats.

alanine metabolism, taurine and hypotaurine metabolism, inositol phosphate metabolism, pyruvate metabolism, GABAergic synapse, and GABA shunt is affected by Mn in MHE (S3 Table) (Fig 4).

This study used [1]H-NMR metabolomics to compare the key metabolites in MHE, chronic hyperammonium (CHA) and chronic hypermanganese (CHM) rat models. Results indicated that abnormal metabolism of glutamine and TCA circle in MHE was affected by the ammonia, and abnormal metabolism of γ-amino butyric acid (GABA) was affected by the manganese (Mn), and abnormal metabolism of glycolysis and branched chain amino acids (BCAAs) metabolism was affected by both ammonia and Mn.

Chronic liver injury in MHE impairs liver detoxification of ammonia and Mn, both of which are neurotoxic and able to cause cognitive impairment [13, 14]. In this study, the blood ammonia was found slight elevated in CHM rats. This may be caused by the liver injury induced by Mn [15, 16]. The results suggest that Mn may have an effect on blood ammonia elevation. However, the brain Mn contents were not significantly changed in CHA rats. Therefore, ammonia may have no effect on brain Mn elevation.

Lactate and alanine are produced from pyruvate and derived in the process of glycolysis [17]. Under normal conditions, the pyruvate is converted to oxaloacetate by pyruvate carboxylase (PC), which is crucial in regulating gluconeogenesis and TCA cycle. Previous studies reported that both ammonia and Mn have been demonstrated to be able to increase glycolysis [18, 19]. The activity of PC was found elevated in CHA, CHM and MHE rats, which may lead to route of pyruvate to lactate production through lactate dehydrogenase. The accumulation of lactate plays an important role in the development of brain edema and cognitive impairment in MHE [20]. Myo-inositol is an osmotic substances efflux from the astrocytes as a compensatory mechanism in brain edema in MHE [21]. Decrease of myo-inositol were both found in hyperammonemia and welders exposed to manganese [18, 22]. Increase lactate and decrease myo-inositol both cause brain edema in MHE.

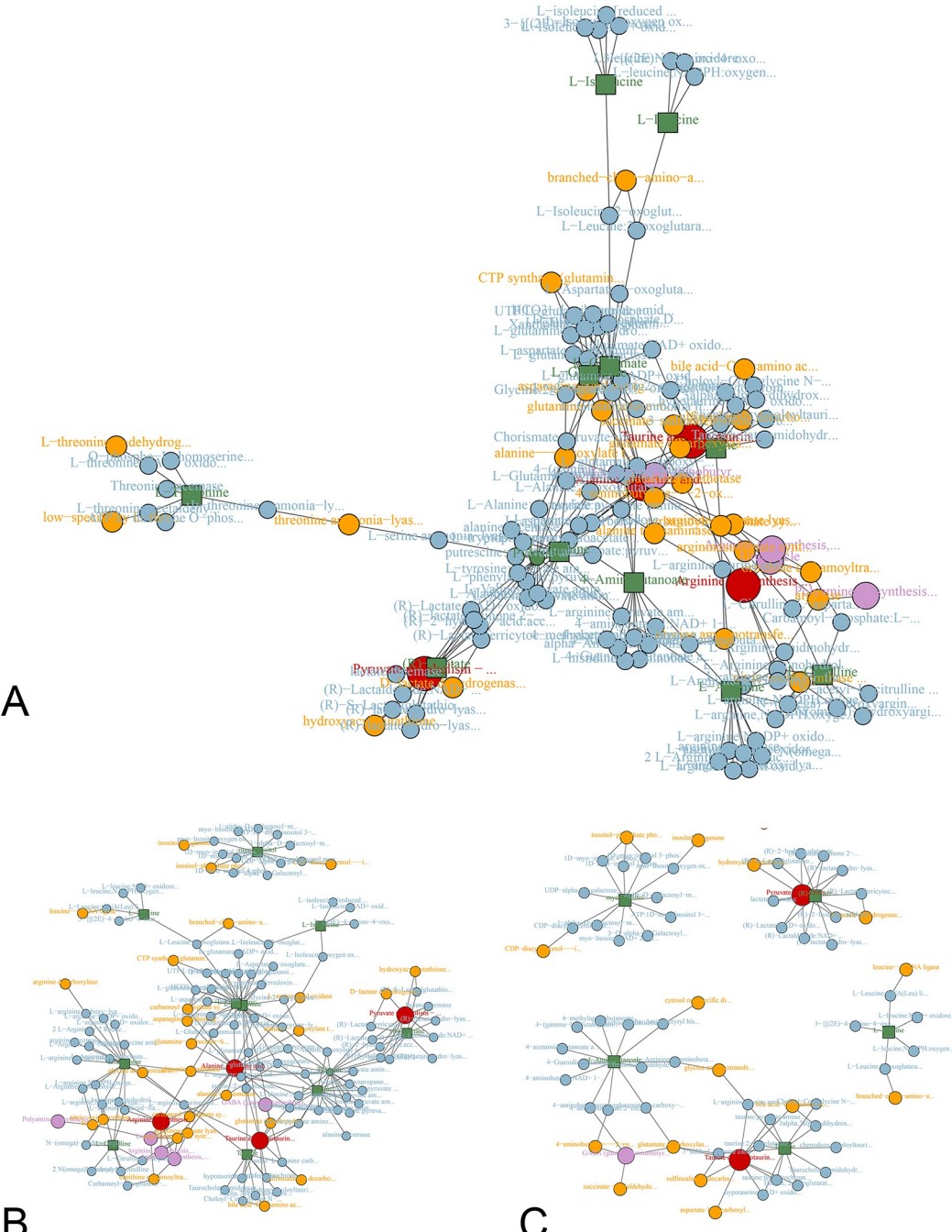

**Fig 4. Metabolic pathways of the key metabolites.** The metabolic pathways involved in the MHE rats (A). The metabolic pathways involved both in the CHA and CHM rats (B). The metabolic pathways involved both in the CHM and MHE rats (C).

Previous study reported the contents of BCAAs in the brain of HE rats decreased while the contents of aromatic amino acids increased [23]. Chronic hyperammonemia rats showed a "high consumption" state of BCAAs, which may be related to the increased metabolism and the conversion of BCAAs to glutamate and glutamine [24]. Results indicated that the decreased BCAAs may be related to the enhancement of brain metabolism caused by both ammonia and

Mn in MHE. Brain accumulation of ammonia and Mn have also shown to reduce the activity of α-ketoglutarate dehydrogenase, a rate-limiting enzyme in the TCA cycle [25]. Impaired aerobic glycolysis and TCA cycle could reduce the ATP production. However, ATP levels remained stable when BCAAs consumption serves as a key source of energy to maintain ATP levels during conditions in impaired TCA cycle [24].

The concentrations of glutamate and glutamine in the striatum of CHA rats were significantly higher than those of CON rats. Glutamate is an important excitatory neurotransmitter in the central nervous system. Though glutamine synthetase (GS), glutamine is converted from glutamate and ammonia, which is an important detoxification way of ammonia in the brain. Under normal circumstances, astrocytes transporters can effectively remove extracellular glutamate and maintain the balance of glutamate/glutamine in the internal environment [26]. High blood ammonia can lead to an increase of glutamate and accumulation of glutamine in the brain, which is another cause of osmotic pressure increasing and astrocytes edema in MHE [27]. The increase of glutamate/glutamine were only observed in CHA rats. As reported in previous studies, a slight decrease rather than increase of glutamate/glutamine was shown in CHM rats [28]. The activity of GS was found elevated in CHA and MHE rats, which may lead to increase of glutamine production. A previous study reported that the inhibitor of GS could reduce the cerebral glutamine of HE rats [29].

Enrichment and pathway analysis indicated that the metabolic changes of GABA are mainly caused by Mn in MHE. However, on the contrary, ammonia showed no effect on GABA metabolism. GABA is the most common inhibitory neurotransmitter and has a great effect on neuronal signal transduction [30]. Previous studies showed that an increase of GABA concentration in the striatum of CHM and MHE rats, which dues to the excessive deposition of Mn resulting a reduction of GABA production and clearance [4].

## Conclusions

Both ammonia and Mn play roles in the abnormal metabolism of MHE. Chronic hypermanganese could lead to elevated blood ammonia. However, chronic hyperammonemia could not lead to brain Mn deposition. Consideration should be given to simultaneously reducing the toxicity of ammonia and Mn in the treatment of MHE.

## Supporting information

**S1 Table. Metabolic pathways of the the key metabolites involved in the CHA and MHE rats.**
(DOCX)

**S2 Table. Metabolic pathways of the the key metabolites involved in the CHM and MHE rats.**
(DOCX)

**S3 Table. Metabolic pathways of the the key metabolites involved in the MHE rats.**
(DOCX)

**S4 Table. The chemical shifts for the 45 metabolites characterized by NMR spectroscopy.**
(DOCX)

**S1 Fig. The full integrated NMR spectrum of the 45 metabolites characterised by NMR spectroscopy.** 1, L-Isoleucine; 2, L-Leucine; 3, L-Valine; 4, alpha-Hydroxyisobutyric Acid; 5, L-Threonine; 6, L-Lactic acid; 7, L-Alanine; 8, L-Lysine; 9, Gamma-Aminobutyric acid; 10, L-Arginine; 11, Acetic Acid; 12, Pyroglutamic Acid; 13, L-Glutamic Acid; 14, L-Glutamine; 15,

Citric acid; 16, L-Aspartic acid; 17, Creatine; 18, Choline; 19, Phosphorylcholine; 20, L-Cysteine; 21, myo-Inositol; 22, Taurine; 23, Ethanolamine; 24, L-Phenylalanine; 25, Glycine; 26, Uracil; 27, Threonic Acid; 28, Guanidoacetic acid; 29, L-Tyrosine; 30, L-Asparagine; 31, Carnosine; 32, Nicotinuric Acid; 33, Guanosine monophosphate; 34, Hypoxanthine; 35, L-Serine; 36, Glycerol; 37, L-Proline; 38, L-Methionine; 39, Citrulline; 40, Malic Acid; 41, Beta-D-Glucose 6-phosphate; 42, Acetylglycine; 43, Argininosuccinic Acid; 44, Methylguanidine; 45, Ornithine.
(TIF)

## Acknowledgments

We thank Chun-Hua Guo from Shanghai Sensichip Infotech Co. Ltd, Shanghai, China in the assistance of metabolomics data analysis.

## Author Contributions

**Conceptualization:** Ying Li, Jin-Wei Qiang.

**Data curation:** Xue-Fei Liu, Jing-Jing Lu, Ying Li, Xiu-Ying Yang.

**Formal analysis:** Xue-Fei Liu, Ying Li.

**Funding acquisition:** Ying Li, Jin-Wei Qiang.

**Investigation:** Xue-Fei Liu, Jing-Jing Lu, Ying Li.

**Writing – original draft:** Xue-Fei Liu, Ying Li.

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
