## [Decision Letter · Decision Letter 0]

1 Jun 2023

PONE-D-23-10343The effects of ammonia and manganese on abnormal metabolism of minimal hepatic encephalopathy: a comparison metabolomics studyPLOS ONE

Dear Dr. LI,

Thank you for submitting your manuscript to PLOS ONE. After careful consideration, we feel that it has merit but does not fully meet PLOS ONE’s publication criteria as it currently stands. Therefore, we invite you to submit a revised version of the manuscript that addresses the points raised during the review process.

We look forward to receiving your revised manuscript.

Kind regards,

Kent Lai

Academic Editor

PLOS ONE

3. We note that this submission includes NMR spectroscopy data. We would recommend that you include the following information in your methods section or as Supporting Information files:

1) The make/source of the NMR instrument used in your study, as well as the magnetic field strength. For each individual experiment, please also list: the nucleus being measured; the sample concentration; the solvent in which the sample is dissolved and if solvent signal suppression was used; the reference standard and the temperature.

2) A list of the chemical shifts for all compounds characterised by NMR spectroscopy, specifying, where relevant: the chemical shift (δ), the multiplicity and the coupling constants (in Hz), for the appropriate nuclei used for assignment.

3)The full integrated NMR spectrum, clearly labelled with the compound name and chemical structure.

We also strongly encourage authors to provide primary NMR data files, in particular for new compounds which have not been characterised in the existing literature. Authors should provide the acquisition data, FID files and processing parameters for each experiment, clearly labelled with the compound name and identifier, as well as a structure file for each provided dataset. See our list of recommended repositories here: https://journals.plos.org/plosone/s/recommended-repositories

Additional Editor Comments:

Your manuscript was reviewed by two experts (physician-scientists to be precise) with extensive knowledge in the fields of metabolism and metabolomics. While both reviewers found your work interesting, they have significant concerns on some of results interpretation and the language used in the text. For instance, Reviewer 1 did not entirely agree with the chosen Title. In addition, it appears to be confusing that metabolism and biosynthesis were reported in the way they were being described. Most importantly, the results came across as descriptive without a clear hypothesis on the mechanisms underlying the pathophysiology. For Reviewer 2, the manuscript could be further improved if the animal models can be better characterized beyond water maze test and be stated on how much they mimic human patients. The same reviewer also asked how the knowledge gained can be translated to better care for the patients, suggesting more mechanistic studies are desired to understand the observed metabolomic data. All in all, I would suggest you and your team to study the comments of the reviewers and when you have sufficient new data to address them, you can consider submitting a revised manuscript for further evaluation.

Reviewers' comments:

Reviewer's Responses to Questions

**Comments to the Author**

1. Is the manuscript technically sound, and do the data support the conclusions?

Reviewer #1: Partly

Reviewer #2: Yes

2. Has the statistical analysis been performed appropriately and rigorously? 

Reviewer #1: I Don't Know

Reviewer #2: Yes

3. Have the authors made all data underlying the findings in their manuscript fully available?

Reviewer #1: Yes

Reviewer #2: Yes

4. Is the manuscript presented in an intelligible fashion and written in standard English?

Reviewer #1: Yes

Reviewer #2: Yes

5. Review Comments to the Author

Reviewer #1: This study aims to investigate the role of high ammonia and of high manganese in minimal hepatic encephalopathy.

The language needs some attention.

The title is somewhat unclear: manganese and ammonia abnormalities are caused by metabolic disturbances resulting from liver dysfunction. They do not affect the metabolism per se. However possibly they may affect each other and play a role in minimal hepatic encephalopathy.

The title mentions ammonia and manganese. The aim of the study only mentions manganese.

I have insufficient expertise to comment on the biochemical measurement methodology

The authors report on the effects of MHE, CHA and CHM on metabolic pathways. Authors write that “The pathway analysis of the thirteen key metabolites from MHE shows that of the arginine biosynthesis, alanine, aspartate and glutamate metabolism, glycine, serine and threonine metabolism, valine, leucine and isoleucine biosynthesis, arginine and proline metabolism, taurine and hypotaurine metabolism, pyruvate metabolism, aminoacyl-tRNA biosynthesis, and GABAergic synapse is evolved in the MHE metabolism

To me it is not clear why sometimes changes in metabolism and sometimes changes in “biosynthesis” are reported. They state that biosynthesis of valine leucine and isoleucine are evolved. How prominent is the biosynthesis of the branch chain aminoacids in rats in a normal state?

The authors state that in MHE mice both arginine biosynthesis and arginine metabolism and other metabolites are affected by ammonia. It is not clear to me on what this is based. Clearly, there is a number of changes in pathways, but I am not sure it can be stated what is the causal effect. A ratio and pathophysiology is lacking.

In the conclusion, indeed it is possible to state that The braim Mn content was increased, perhaps causing impaired ammonia metabolism, and the chronic hyperammonemia did not cause Mn deposition.

However, I am not sure it can be stated so strongly that The metabolism of related to the glycolysis and TCA circle in MHE is mainly caused by ammonia, while that of GABA abnormal metabolism is mainly caused by Mn.

Reviewer #2: This study used 1H-NMR metabolomics to investigate the effects of ammonia and manganese in the metabolism of MHE. What they found may help improve the quality of life of patients with MHE in the future.

Major comments

1. What were the causes of chronic hypermanganes?

2. What were the phenotypes of the “MHE rats”? Were there any changes in liver function and histology in the “MHE rats”? Whether the “MHE rats” described in the paper can mimic the characteristic of patients with MHE?

3. The sentences “The blood ammonia was found slight elevated in CHM rats, which may be caused by the liver injury induced by Mn.15,16 The brain Mn contents in CHM rats and MHE rats were significantly higher than those in SOC rats. However, no significant change in brain Mn content were shown in CHA.” (Page 16 Line 6-12) seemed to be controversial with the sentence “Therefore, the results suggest that Mn may have an effect on ammonia metabolism, but ammonia has no effect on brain Mn elevation in MHE”. Please clarify it.

4. What can clinicians learn from the paper? Please amplify the significance of the study.

Minor comments

1. Use the full name when it first appears, eg. Page 1 Line 14, Line 22.

2. Use punctuation appropriately, eg. Page 3 Line 22.

3. It should use “which” rather than “that” in Page 4 Line 1.

4. It should use “sacrificed” rather than “killed” in Page 6 Line 4.

5. Keep the format consistent, eg. Page 6 Line 11, Line 19.

6. PLOS authors have the option to publish the peer review history of their article (what does this mean?). If published, this will include your full peer review and any attached files.

Reviewer #1: No

Reviewer #2: No

---

## [Author Response · Author response to Decision Letter 0]

27 Jun 2023

Dear Editors and Reviewers:

Thanks for the review of our manuscript and thanks for the suggestions. Those comments are valuable and helpful for revising and improving our paper. We have considered carefully these comments and have made relevant corrections point- by-point.

Response: Thanks for the comment. The manuscript was reformatted according to the requirements.

2. Please note that PLOS ONE has specific guidelines on code sharing for submissions in which author-generated code underpins the findings in the manuscript. 

Response: Thanks for the comment. The codes used for statistical analysis in this study can be found at “https://gitee.com/dr_yingli/mn-mrs/tree/master/”.

3. We note that this submission includes NMR spectroscopy data. We would recommend that you include the following information in your methods section or as Supporting Information files:

1) The make/source of the NMR instrument used in your study, as well as the magnetic field strength. For each individual experiment, please also list: the nucleus being measured; the sample concentration; the solvent in which the sample is dissolved and if solvent signal suppression was used; the reference standard and the temperature.

Response: Thanks for the suggestion. More information about the NMR instrument and experimental protocol were provided in the methods section.

2) A list of the chemical shifts for all compounds characterized by NMR spectroscopy, specifying, where relevant: the chemical shift (δ), the multiplicity and the coupling constants (in Hz), for the appropriate nuclei used for assignment.

Response: Thanks for the suggestion. The list was uploaded as supporting information (Supplementary Table 4. The chemical shifts and chemical structure for the 45 metabolites characterized by NMR spectroscopy).

3)The full integrated NMR spectrum, clearly labeled with the compound name and chemical structure.

Response: Thanks for the suggestion. The NMR spectrum was uploaded as supporting information (Supplementary Figure 1. The full integrated NMR spectrum of the 45 metabolites characterized by NMR spectroscopy.).

4. We note that the grant information you provided in the ‘Funding Information’ and ‘Financial Disclosure’ sections do not match. When you resubmit, please ensure that you provide the correct grant numbers for the awards you received for your study in the ‘Funding Information’ section.

Response: Thank you for the reminding. The “Financial Disclosure” was updated.

Response: Thank you for the reminding.

Additional Editor Comments:

Your manuscript was reviewed by two experts (physician-scientists to be precise) with extensive knowledge in the fields of metabolism and metabolomics. While both reviewers found your work interesting, they have significant concerns on some of results interpretation and the language used in the text. For instance, Reviewer 1 did not entirely agree with the chosen Title. In addition, it appears to be confusing that metabolism and biosynthesis were reported in the way they were being described. Most importantly, the results came across as descriptive without a clear hypothesis on the mechanisms underlying the pathophysiology. For Reviewer 2, the manuscript could be further improved if the animal models can be better characterized beyond water maze test and be stated on how much they mimic human patients. The same reviewer also asked how the knowledge gained can be translated to better care for the patients, suggesting more mechanistic studies are desired to understand the observed metabolomic data. All in all, I would suggest you and your team to study the comments of the reviewers and when you have sufficient new data to address them, you can consider submitting a revised manuscript for further evaluation.

Response: Thank you for the comment. 

#1 The title was revised to “The interaction of ammonia and manganese in abnormal metabolism of minimal hepatic encephalopathy: a comparison metabolomics study”

#2 We kept the format consistent by using “metabolism” instead of “biosynthesis”.

#3 A clear hypothesis was state in the in introduction “We assumed that both ammonia and Mn play a role in the abnormal metabolism of MHE with interactions between these two toxic substances”.

#4 We performed narrow beam walking test to evaluate the locomotion of the rats. The results were added in the text and Table 1.

#5 The activities of the enzymes related to the metabolism of lactate, branched chain amino acids, glutamate and glutamine were measured. The results were added in the text and Table 1. And the results were also discussed in the Discussion section.

Reviewers' comments:

Reviewer #1: This study aims to investigate the role of high ammonia and of high manganese in minimal hepatic encephalopathy.

The language needs some attention.

Response: Thank you for the comment. The language of the manuscript was polished.

The title is somewhat unclear: manganese and ammonia abnormalities are caused by metabolic disturbances resulting from liver dysfunction. They do not affect the metabolism per se. However possibly they may affect each other and play a role in minimal hepatic encephalopathy.

The title mentions ammonia and manganese. The aim of the study only mentions manganese.

Response: Thank you for the comment. The study aimed to investigate whether ammonia and manganese may affect each other in the abnormal metabolism of minimal hepatic encephalopathy. We changed the title to “The interaction of ammonia and manganese in abnormal metabolism of minimal hepatic encephalopathy: a comparison metabolomics study” for clarity.

The authors report on the effects of MHE, CHA and CHM on metabolic pathways. Authors write that “The pathway analysis of the thirteen key metabolites from MHE shows that of the arginine biosynthesis, alanine, aspartate and glutamate metabolism, glycine, serine and threonine metabolism, valine, leucine and isoleucine biosynthesis, arginine and proline metabolism, taurine and hypotaurine metabolism, pyruvate metabolism, aminoacyl-tRNA biosynthesis, and GABAergic synapse is evolved in the MHE metabolism

To me it is not clear why sometimes changes in metabolism and sometimes changes in “biosynthesis” are reported. They state that biosynthesis of valine leucine and isoleucine are evolved. How prominent is the biosynthesis of the branch chain aminoacids in rats in a normal state?

Response: Thank you for the inquiry. The pathway names were directly derived from KEGG database. We do agree that using both “biosynthesis” and “metabolism” may cause confusion for readers. We kept the format consistent by using “metabolism” instead of “biosynthesis”. The key metabolites of MHE, CHA and CHM rats were derived by comparing with the control rats. Results showed that the metabolisms of branch chain aminoacids were decreased in MHE, CHA and CHM rats, which indicated that both ammonia and manganese affect the metabolism of branch chain aminoacids.

The authors state that in MHE mice both arginine biosynthesis and arginine metabolism and other metabolites are affected by ammonia. It is not clear to me on what this is based. Clearly, there is a number of changes in pathways, but I am not sure it can be stated what is the causal effect. A ratio and pathophysiology is lacking.

Response: Thank you for the inquiry. The key metabolites were derived by comparing with the controls. The results of enrichment and pathway analysis were derived from further statistic analysis. Results suggested that metabolic pathways may be impaired on considering these abnormal key metabolites.

In the conclusion, indeed it is possible to state that The brain Mn content was increased, perhaps causing impaired ammonia metabolism, and the chronic hyperammonemia did not cause Mn deposition.

However, I am not sure it can be stated so strongly that The metabolism of related to the glycolysis and TCA circle in MHE is mainly caused by ammonia, while that of GABA abnormal metabolism is mainly caused by Mn.

Response: Thank you for the concern. Although, results of enrichment and pathway analysis suggested that glycolysis and TCA circle metabolisms are the main abnormal metabolisms in rat models induced by hyperammonemia, while GABA metabolism is the main abnormal metabolism in rat models induced by hypermanganese. We do agree that it’s not rigorous to state so, on considering that the metabolic pathways were derived from statistic analyzing of key metabolites rather than direct measurement. Thus, we revised the conclusion to “Both ammonia and Mn play roles in the abnormal metabolism of MHE. Chronic hypermanganese could lead to elevated blood ammonia. However, chronic hyperammonemia could not lead to brain Mn deposition. Consideration should be given to simultaneously reducing the toxicity of ammonia and Mn in the treatment of MHE.”.

Reviewer #2: This study used 1H-NMR metabolomics to investigate the effects of ammonia and manganese in the metabolism of MHE. What they found may help improve the quality of life of patients with MHE in the future.

Major comments

1.What were the causes of chronic hypermanganes?

Response: Thank you for the inquiry. The hypermanganes is cased by portosystemic shunting. We clarify it in the introduction “Furthermore, elevated blood manganese (Mn) and brain Mn deposition were reported, due to the presence of portosystemic shunt and bypassing of liver excretion of Mn ”.

2.What were the phenotypes of the “MHE rats”? Were there any changes in liver function and histology in the “MHE rats”? Whether the “MHE rats” described in the paper can mimic the characteristic of patients with MHE?

Response: Thank you for the inquiry. There are three types of MHE. Type A (caused by acute liver dysfunction), type B MHE (caused by portosystemic shunting with none [usually] or mild liver dysfunction) and type C (caused by both liver dysfunction and portosystemic shunting). Type C MHE is the most common type seen in clinical practice (cirrhosis e.g.). We added this in the introduction to make clarity for the readers. MHE is characteristic with no typical clinical manifestations but abnormal in the neurological function assessment. The rats model used in this study was a common model to mimic type C MHE (liver cirrhosis induced by thioacetamide). The MHE rats model used in this study can mimic the characteristic of patients with MHE. Impaired neurological behavior of MHE rat can be assessed using Morris water maze. We also performed narrow beam walking test to evaluate the locomotion of the rats. The results were added in the text and Table 1.

3.The sentences “The blood ammonia was found slight elevated in CHM rats, which may be caused by the liver injury induced by Mn.15,16 The brain Mn contents in CHM rats and MHE rats were significantly higher than those in SOC rats. However, no significant change in brain Mn content were shown in CHA.” (Page 16 Line 6-12) seemed to be controversial with the sentence “Therefore, the results suggest that Mn may have an effect on ammonia metabolism, but ammonia has no effect on brain Mn elevation in MHE”. Please clarify it.

Response: Thank you for the concern. The blood ammonia was found slight elevated in CHM rats, which may be caused by the liver injury induced by Mn. The results suggest that Mn may have an effect on blood ammonia elevation. However, the brain Mn contents were not significantly changed in CHA rats. Therefore, ammonia may have no effect on brain Mn elevation. We revised the sentence for clarity int the text.

4.What can clinicians learn from the paper? Please amplify the significance of the study.

Response: Thanks for the suggestion. Both ammonia and Mn play roles in the abnormal metabolism of MHE. Consideration should be given to simultaneously reducing the toxicity of ammonia and Mn in the treatment of MHE. We added it in the text.

Minor comments

1.Use the full name when it first appears, eg. Page 1 Line 14, Line 22.

Response: Thanks for the suggestion. We revised the abstract according to the suggestions.

2.Use punctuation appropriately, eg. Page 3 Line 22.

Response: Thanks for the suggestion. It was revised in the text.

3.It should use “which” rather than “that” in Page 4 Line 1.

Response: Thanks for the suggestion. We changed “that” to “which” in the text.

4.It should use “sacrificed” rather than “killed” in Page 6 Line 4.

Response: Thanks for the suggestion. We changed “killed” to “sacrificed” in the text.

5.Keep the format consistent, eg. Page 6 Line 11, Line 19.

Response: Thanks for the suggestion. We changed “NMR” to “1H-NMR” in the text.

---

## [Decision Letter · Decision Letter 1]

25 Jul 2023

The interaction of ammonia and manganese in abnormal metabolism of minimal hepatic encephalopathy: a comparison metabolomics study

PONE-D-23-10343R1

Dear Dr. Li,

We’re pleased to inform you that your manuscript has been judged scientifically suitable for publication and will be formally accepted for publication once it meets all outstanding technical requirements.

Kind regards,

Kent Lai

Academic Editor

PLOS ONE

Additional Editor Comments (optional):

Reviewers' comments:

Reviewer's Responses to Questions

**Comments to the Author**

1. If the authors have adequately addressed your comments raised in a previous round of review and you feel that this manuscript is now acceptable for publication, you may indicate that here to bypass the “Comments to the Author” section, enter your conflict of interest statement in the “Confidential to Editor” section, and submit your "Accept" recommendation.

Reviewer #1: All comments have been addressed

Reviewer #2: All comments have been addressed

2. Is the manuscript technically sound, and do the data support the conclusions?

Reviewer #1: Yes

Reviewer #2: Yes

3. Has the statistical analysis been performed appropriately and rigorously? 

Reviewer #1: I Don't Know

Reviewer #2: Yes

4. Have the authors made all data underlying the findings in their manuscript fully available?

Reviewer #1: Yes

Reviewer #2: Yes

5. Is the manuscript presented in an intelligible fashion and written in standard English?

Reviewer #1: Yes

Reviewer #2: Yes

6. Review Comments to the Author

Reviewer #1: (No Response)

Reviewer #2: (No Response)

7. PLOS authors have the option to publish the peer review history of their article (what does this mean?). If published, this will include your full peer review and any attached files.

Reviewer #1: No

Reviewer #2: No

---

## [Editor Report · Acceptance letter]

27 Jul 2023

PONE-D-23-10343R1 

The interaction of ammonia and manganese in abnormal metabolism of minimal hepatic encephalopathy: a comparison metabolomics study 

Dear Dr. Li:

I'm pleased to inform you that your manuscript has been deemed suitable for publication in PLOS ONE. Congratulations! Your manuscript is now with our production department. 

Kind regards, 

on behalf of

Dr. Kent Lai 

Academic Editor

PLOS ONE